# Advanced paramedics' restraint decision-making when managing acute behavioural disturbance (ABD) in the UK pre-hospital ambulance setting: A qualitative investigation

Jaqualine Lindridge[1,2]*, Timothy Edwards[2], Leda Blackwood[3]

1 Department for Health, University of Bath, Bath, Somerset, England, 2 London Ambulance Service NHS Trust, London, England, 3 Department of Psychology, University of Bath, Bath, Somerset, England

* j.lindridge@bath.ac.uk

**Data Availability Statement:** Data are available via the University of Bath Research Data Archive and are available to researchers on application via:

## Abstract

Acute behavioural disturbance (ABD), sometimes called 'excited delirium', is a medical emergency. In the UK, some patients presenting with ABD are managed by advanced paramedics (APs), however little is known about how APs make restraint decisions. The aim of this research is to explore the decisions made by APs when managing restraint in the context of ABD, in the UK pre-hospital ambulance setting. Seven semi-structured interviews were undertaken with APs. All participants were experienced APs with post-registration, post-graduate advanced practice education and qualifications. The resulting data were analysed using reflexive thematic analysis, informed by critical realism. We identified four interconnected themes from the interview data. Firstly, managing complexity and ambiguity in relation to identifying ABD patients and determining appropriate treatment plans. Secondly, feeling vulnerable to professional consequences from patients deteriorating whilst in the care of APs. Thirdly, negotiating with other professionals who have different roles and priorities. Finally, establishing primacy of care in relation to incidents which involve police officers and other professionals. A key influence was the need to characterise incidents as medical, as an enabler to establishing clinical leadership and decision-making control. APs focused on de-escalation techniques and sought to reduce physical restraint, intervening with pharmacological interventions if necessary to achieve this. The social relationships and interactions with patients and other professionals at the scene were key to success. Decisions are a source of anxiety, with fears of professional detriment accompanying poor patient outcomes. Our results indicate that APs would benefit from education and development specifically in relation to making ABD decisions, acknowledging the context of inter-professional relationships and the potential for competing and conflicting priorities. A focus on joint, high-fidelity training with the police may be a helpful intervention.

Lindridge, J., Edwards, T., Blackwood, L., 2024. Dataset for "Advanced paramedics' restraint decision-making when managing acute behavioural disturbance (ABD): A UK based qualitative investigation.". Bath: University of Bath Research Data Archive. https://doi.org/10.15125/BATH-01326.

**Funding:** The author(s) received no specific funding for this work.

**Competing interests:** The authors have declared that no competing interests exist.

# Introduction and background

Acute behavioural disturbance (ABD) is a term used to describe a state of significant agitation or disordered behaviour, with or without physiological compromise [1] and is the term most widely used by professionals in the UK [1–3]. Restraint and the management of ABD is a controversial area of policy and practice internationally [4, 5]. Whilst physical restraint tends to be the preserve of the police, paramedics in some areas administer pharmacological restraint in the form of ketamine [6], anti-psychotics and/ or benzodiazepines to patients presenting with agitation [7]. Sadly, some deaths in the context of ABD have been linked to both physical restraint delivered by law enforcement professionals [8, 9] and pharmacological restraint delivered by paramedics in the US [10]. In the UK, the role of some paramedics in restraint decision-making is evolving, with the recent introduction of pharmacological restraint in some Emergency Medical Services (EMS) [11], following the development of advanced paramedics (APs) and growing concerns in this area of practice. ABD patients typically present initially to police services in public spaces or in police custody [12]. APs usually attend later, often at the request of the police themselves. APs may arrive at a chaotic scene [13] and make restraint decisions, although they will not normally be responsible for carrying out physical restraint. Little is known about how APs make restraint decisions.

A narrative literature review was undertaken to identify studies which could shed light on what might influence APs when determining the need for restraint in the context of ABD. Studies relevant to decision-making in two key contexts were sought. Firstly, the restraint of patients experiencing ABD, or a similar problem, in any clinical practice setting or by any healthcare professional group, and secondly, paramedic and AP decision-making more generally. To clarify the difference between paramedics and APs, in the UK, paramedics are registered healthcare professionals who work autonomously within a defined scope of practice. APs are experienced paramedics, with post-registration, post-graduate advanced practice qualifications who within some UK EMS providers are selectively tasked to suspected ABD incidents to provide senior clinical leadership and, where indicated, a pharmacological intervention which is outside of the scope of practice of general paramedics. Whilst APs have an advanced scope of practice and educational background, they remain paramedics by profession and research relating to the paramedic profession was also sought for this reason. Throughout this paper, clinicians are defined as skilled and qualified healthcare workers. The term 'ambulance clinician' is a generic label which typically refers to a paramedic or qualified emergency medical technician (EMT) working in the UK NHS ambulance setting.

Research has identified that paramedics have felt unprepared by their formal training to manage behavioural and mental health emergencies and tend to rely on tacit knowledge and their general professional experience [14, 15], it is not known if this extends to paramedics working at an advanced practice level. Tacit knowledge has been described as "...*knowledge-in-practice developed from direct experience and action; highly pragmatic and situation specific; subconsciously understood and applied; difficult to articulate; usually shared through interactive conversation and shared experience.*" [16]. Furthermore, restraint-related research in the in-patient psychiatric setting has highlighted that other healthcare professionals rely on tacit knowledge when making restraint decisions [17], an issue which may be particularly relevant to APs whose scope of practice has been recently developed to include pharmacological restraint [11].

Research on paramedic decision-making more broadly finds professional vulnerability and a fear of repercussions is a dominant theme with paramedics fearing disciplinary investigation and litigation in relation to managing patients who self-harm [14]; making non-conveyance decisions [18–21]; and cardiac arrest management [22, 23]. More generally, there is anxiety

around practice in what has been described as a 'hostile environment' of discipline and blame within UK ambulance services [24]. Paramedics also fear a lack of support from their employers in a range of circumstances [20, 25]. Although some of these studies included paramedics working at a specialist or enhanced level [20–22, 25], it is unclear whether and how transition to AP status affects professional fears and vulnerability. Some studies suggest that decisions in the ambulance practice arena may be more complicated than treatment guidelines account for [20, 26], with guidelines failing to respond to the degree of ambiguity and variation which ambulance clinicians encounter in practice. In a similar vein, access to limited information has been identified as a barrier to decision-making in cardiac arrest management [27] and end of life care [28]. This suggests that the decision-making landscape in which APs operate is challenging.

Current thinking is that paramedic decision-making is based on atomised individual-level 'rational' processes, with studies focusing on cognitive processing [29–31]. However, decisions made in relation to pre-hospital restraint take place in a complex social environment which includes bystanders, other professionals, such as the police, and other ambulance clinicians. The context in which these decisions are made is important. Brandling *et al.* observed the influence of police officers on clinical decisions in the context of cardiac arrest [23], with ambulance clinicians (including paramedics) finding it difficult to overturn police decisions in relation to whether to start resuscitation. Other studies have also acknowledged the pressure felt from bystanders or others at the scene. Burrell *et al.* found that ambulance clinicians encountered pressure from bystanders when responding to patients experiencing seizures in the public place, particularly in respect of providing pharmacological interventions to control seizure activity [18]. Murphy-Jones and Timmons reported that ambulance clinicians experienced pressure from family, care home staff and other paramedics to convey patients at the end of their life to hospital [28], even where that appeared contrary to the wishes of the patient. Issues such as these are likely to be relevant to the experience of APs arriving at scenes with access to additional medicines outside of the scope of practice of general paramedics.

Several studies report that nurses consider restraint to be a necessary, but undesirable activity that is contrary to their caring role [17, 32, 33], however little is known on the views of APs in making restraint decisions, and no studies specifically relating to APs were identified.

How APs make restraint decisions in the context of ABD is an important topic with real-world implications for patients and professionals [34]. Current evidence suggests decisions may be complex [20, 22, 26], and a deeper understanding may be beneficial in developing care in this area of practice. Whilst there has been research undertaken on restraint in other professions [32], there is a lack of research relating to APs as a specific professional group working in the clinical setting of interest, the pre-hospital ambulance setting.

The aim of this study is to explore the decisions made by APs when managing a restraint incident in the context of ABD, and to identify opportunities to improve performance in this area of practice. Recognising the variation in terminology relating to pharmacological interventions in ABD, the term 'chemical sedation' will be used throughout this report to describe the administration of sedative or anti-psychotic agents as it is the term most frequently used by those undertaking the intervention.

## Study design and methods

### Research philosophy

This is an exploratory, qualitative study of APs practicing in a large, metropolitan UK emergency medical service (EMS). This study was conducted from a critical realist standpoint, following Campbell [35] and Maxwell's [36] approach which combines ontological realism and

epistemological constructivism. Ontological realists believe that reality, and the components of reality, exist independently of consciousness [37], and epistemological constructivists believe that our understanding of reality is a construction based on human perception [36]. An alternative to positivism developed by Roy Bhaskar, critical realism was summed up by Sturgiss and Clark as stating that the '...*evidence we observe can come close to reality but is always a fallible, social and subjective account of reality.*" [38] Here, we take the position that the phenomenon of decision-making in the context of ABD is real, and that APs make sense of this phenomenon in a subjective and socially constructed way.

## Research context

London Ambulance Service National Health Service (NHS) Trust (LAS), provides free at-the-point-of-access emergency medical and ambulance services across London, a large metropolitan city in the UK. Accessed via a single emergency telephone number, contacts are triaged using the Medical Priority Dispatch System (MPDS™). The typical response to a high-acuity emergency is the attendance of an emergency ambulance, staffed by a paramedic and emergency medical technician, with or without the additional attendance of a solo paramedic attending by car or motorcycle. LAS has a well-established critical care advanced practice service, and additionally deploys APs to suspected cases of ABD, with a recent study identifying that 237 patients were coded as presenting with ABD by APs in a 12-month period [13]. APs are permitted to administer chemical sedation to patients using Patient Group Directions (PGDs), a legal framework which enables registered health care professionals to administer parenteral medicines without a prescription in the UK.

## Sampling and recruitment

The study was advertised using internal bulletins, social media, and word of mouth. Seven APs were recruited using convenience sampling. Overall, approximately 18% of the relevant study population participated. Participants were recruited between May and November 2021.

There is no consensus on the sample size required for qualitative studies [39]. Braun and Clarke suggest that researchers "...*make an in-situ decision about the final sample size, shaped by adequacy (richness, complexity) of the data for addressing the research question...*" [40]. Our research focussed on a specific area of clinical practice and the "...*experiences, knowledge, or properties...*" of our participants were highly specific in relation to it as key informants [37, 40, 41]. All participants were able to discuss several cases each from their personal experience in depth, producing good quality interview dialogue.

## Data collection and processing

Seven semi-structured, in-depth interviews were undertaken, lasting between 52 and 66 minutes. The interviews examined participants' experiences by enabling them to tell stories about ABD incidents they had attended. Interviews were conducted with the least structure and interviewer input as practicable to allow flexibility for interviewees to lead the discussion and talk about what was meaningful for them and for the interviewer to be flexible in eliciting new information [42]. Participants were asked to tell the interviewer about a time they had to decide whether to restrain a person with ABD or not, and probes were used to explore the interactions between those at scene, how participants felt about managing ABD, and why they chose to manage the cases in the way that they did. APs are often natural story tellers who frequently engage in reflective practice and often describe the incidents they attend in peer-to-peer conversations, this was evident in interviews with all participants speaking freely and in depth about their experiences. As a paramedic themselves, the interviewer was an insider researcher and

therefore shared a degree of professional experience, identity and language with participants [43]. Insider research benefits from early acceptance of the interviewer and can facilitate a level of openness and access to data which can be difficult to obtain from outsider research [44, 45]. The conversations were acquaintance interviews which enabled a good rapport to be established quickly [44]. It was also recognised that there is risk in insider research for interviewees to assume similarity with researchers and limit their explanations, and for researchers to confuse their own perceptions with those of participants [44]. To avoid this, informal member checking took place to confirm understanding of participants' points. This was achieved by the interviewer summarising and restating points and inviting the participant to confirm accuracy.

Interviews were conducted and digitally recorded using Microsoft Teams, before being transcribed intelligent verbatim. The seven interviews generated 234 pages of data, consisting of 65,246 words. Each interview produced rich and complex data with a high level of depth. Based on the specificity of the participants and the quality of interview dialogue, these interviews were deemed to have generated sufficient relevant data to enable the development of meaningful themes [40]. All data were stored on managed and secure computer servers. Authors JL and LB had access to information which could identify participants during data collection, up to the point of anonymisation. Author TE had access to anonymised data only, as access to participant personal information was not required for both supervisors.

## Data analysis

Braun and Clarke's [46] method of reflexive thematic analysis (RTA) was used throughout the study. RTA is a method of theoretically flexible analysis which uses systematic coding of data to identify and interpret patterns and develop themes and acknowledges the part played by researchers in knowledge production. RTA values "*. . .a subjective, situated, aware and questioning researcher. . .*" [46]. The three authors involved in the analysis were the Chief Investigator who is a woman consultant paramedic with an interest in medical ethics, professional duty and patient safety; a man consultant paramedic with extensive experience in developing and providing advanced paramedic critical care practice; and a senior woman social psychologist who conducts research on contextually situated social interactions and decision-making. The different perspectives and subjectivities of the co-authors were examined through dialogue throughout data analysis, as part of researcher reflectivity. Throughout the analysis, the Chief Investigator made memos noting their reflections and interpretations of the data, situated within their own context and view of the topic.

We used RTA inductively, with an emphasis on broad thematic patterning. Data analysis began following the first interview, and followed an exploratory and iterative approach [47]. The CI first familiarised themself with the data through listening to recordings during transcription and repeated reading of transcripts. This was followed by coding and theming stages; these stages involved all authors in sharing and discussing interpretations and returning to the data. The development of themes and an analytic narrative was a collaborative process [48].

## Ethical approval

Ethical approval for this study was provided by the Heath Research Authority (IRAS: 265010).

## Results

### Participants

Of the seven participants, one was female and six were male, noting that most APs in the study population were male at the time of the research. To protect anonymity, gender-neutral

pseudonym names have been used. All participants were experienced APs with clinical education at the post graduate level and were adults of working age. Within the study site, it is part of the scope of practice for APs specialising in critical care to administer ketamine (an NMDA receptor antagonist used for sedation), haloperidol (an antipsychotic) and/ or midazolam (a benzodiazepine) to patients experiencing ABD, autonomously and in accordance with relevant PGDs.

## Findings

We identified four themes from the dataset. The first, managing complexity and ambiguity, encompassed difficulties in differentiating ABD from other differential diagnoses and determining the right clinical management plan, including use of chemical sedation. The second theme, feeling vulnerable to professional consequences, related to participants' concerns regarding the high-risk nature of the syndrome of ABD, and the potential for professional repercussions if cases led to poor patient outcomes. The third theme, recognising and establishing primacy of care, encompassed making leadership and accountability connections between professionals and the patient. The fourth, and final theme, needing to negotiate with other professionals, related to the interactions between APs and other clinicians or police officers at the incident scene and APs' perceptions of achieving agreement on how to manage ABD patients. These four themes are reported in the following section, with participants' data extracts presented alongside the associated analytic narrative.

**Managing complexity and ambiguity.** The theme of managing complexity and ambiguity encompassed challenges with differentiating ABD from other forms of behavioural agitation and weighing up what to do, including identifying the right pharmacological intervention for patients who required chemical sedation.

For example, 'Eddie' described trying to differentiate which patients presenting with agitation were potentially experiencing ABD, to determine how they should be managed clinically.

> *"So I try to engage with the male, [. . .] and it became clear that although the male was having some paranoid thoughts, [. . .] he would have little moments where he would engage with me, and look at me and actually would say something sensible."*

'Eddie'

Here, 'Eddie' sheds light on an important differentiating sign of ABD, the ability to 'engage' with the patient and establish if they have a loss of insight. Other participants also referred to the ability to 'engage' with ABD patients as a differentiating sign, such as 'Jordan' who described assessing how a patient connected with his appearance during one interaction: *". . .he knows what he's saying, he reckons I've got a big [. . .] your brain is working. . .".* There is ambiguity in differentiating ABD from other forms of agitation which APs must navigate to make treatment decisions. Participants appeared to find the ability to engage with patients helpful in reducing this uncertainty.

As APs with chemical sedation within their scope of practice, many of the participants were particularly concerned with the decisions they made in relation to this intervention. 'Pat' described a case where they used chemical sedation to manage an ABD patient, and this acted as an enabler to good care.

> *"We had control, literally control of the scene and were able to [. . .] assess him properly. Let's do a top to toe, check for any injuries, have we missed anything, all the things you'd like to do with these patients, but you can't 'cause you can't get near them."*

'Pat'

The use of chemical sedation here allows 'Pat' to *'literally [take] control'* and creates an opportunity to examine the patient. Prior to the chemical sedation taking effect, the police were in effect controlling the scene, and the combination of agitation and restraint prevented clinical assessment, prolonging the lack of information in relation to the patient's physiological condition, increasing the complexity of the scenario and heightening the clinical risk. The use of chemical sedation has allowed access to the patient, and thus access to information which in turns begins to reduce the complexity and ambiguity of the incident and assists in determining subsequent ongoing management.

In making restraint decisions, 'Charlie' spoke about their concerns in relation to how much information they had about patients' underlying physiology and how this affected their choices, particularly in relation to chemical sedation.

> *"What you are doing is [. . .] taking an action, be that physical restraint or chemical sedation to a patient who you've got no idea what their baseline physiology is, and you've therefore got no idea how they will respond to those agents."*

'Charlie'

'Charlie' is particularly concerned about the unpredictability of the patient's clinical course which results from a lack of information, and the increased risks which result. 'Jordan' was also concerned about this issue, adding that *". . .you just don't know how people respond, sometimes you give somebody 5 mg of midazolam, and they're GCS 3 [unresponsive to verbal and physical stimulus] and need airway management, but sometimes people take 15 mg of that stuff, and they're still really agitated, really aggressive and fighting the police."* There is an inherent uncertainty here about the outcome of '*taking an action*' which is an important context for decision-making. APs are making decisions which have serious consequences and are doing so in the knowledge that they have a limited influence of the outcomes.

The above accounts speak to ambiguity of diagnosis. We end with 'Charlie' who describes a case in which they were dealing with legal ambiguities and the need to make a defensible decision.

> *"There is a level of immediacy, there is no least restrictive option and [. . .] the Mental Capacity Act gives us a basis of action [. . .] and we are acting in his best interests."*

'Charlie'

In this example, 'Charlie' has used a systematic and structured approach to work through relevant issues to their decision. This appears to have simplified the decision and reassured 'Charlie' that there is a legal basis for restraint, reducing ambiguity.

**Feeling vulnerable to professional consequences.** The feeling vulnerable to professional consequences theme incorporates the concerns which participants spoke of in relation to the high-risk nature of these incidents, and what that might mean for professionals should the patient have a poor outcome, in particular concerns relating to medicolegal processes.

In the first example in this theme, 'Eddie' recalls a case from early in their advanced practice career where they used chemical sedation, and the patient stopped breathing for a time immediately after the drugs were administered.

> *"And to this day I can still remember exactly what happened in that case. Because for those few seconds where it looked like she'd stopped breathing, that was absolutely terrifying, because my immediate thought was—I've done this. By sedating the patient, and have I given*

*the right drugs, have I made a drug error, have I given too much? And actually, I double checked, and triple checked my drug dosages prior to administering the drugs. And still I thought I may have made an error, but I hadn't."*

'Eddie'

'Eddie' is fearful of the patient coming to harm and their first thought was that they had "*killed the patient*". There is an emphasis here on 'Eddie's' actions potentially causing harm to the patient, and a fear so impactful they can recall the incident in detail many years later. 'Eddie' appears to focus initially on the potential for personal error, rather than other potential sources of the patient's apnoea. They express fear and doubt, even though they know that they have followed the correct procedures.

'Eddie' also reflected on how similar procedures are managed in different healthcare services and added that unlike other clinical settings APs work in relative isolation when chemical sedation is concerned. Whilst a team of ambulance professionals work together to care for the patient, the AP is typically the only clinician there with specialist knowledge and experience of the procedure, concentrating the responsibility on their shoulders—as 'Eddie' said, "*this is on me.*" This perhaps contributes to a sense of professional vulnerability due to a lack of immediate shared decision-making with other clinicians.

'Pat' was also concerned about managing ABD cases in terms of providing chemical sedation, particularly around the risk of deterioration and how APs would experience associated investigations.

*"ABD terrifies me. [. . .]. We know at some point we are going to sedate somebody who will die. And that may be as a result of the sedation or [. . .] they were going to die anyway. But we will never know that. And I fear being involved in that I really do. [. . .] There's internal investigations to go through. Then there's an external process. There's probably a referral to HCPC. [. . .] Thinking that someone at some point is going to uncover something you should or shouldn't have done. [. . .] It sounds terrible, but this, this could be my last ever job. You know, this could be the job that breaks me."*

'Pat'

There are two key issues here, a fear of a patient dying and a fear around the investigatory processes which would follow that. Firstly, like 'Eddie', 'Pat' is '*terrified*' and expresses '*fear*' about a patient dying. In saying '*we will never know*' we see that there is an intrinsic ambiguity in causes of deterioration, and this was an important burden for participants. Secondly, the idea of a death creates great apprehension for 'Pat', who refers to multiple investigations following such an eventuality. 'Pat' also expects a summary or arbitrary fitness to practice referral to the Health and Care Professions Council (HCPC), the regulatory body for paramedics in the United Kingdom and anticipates considerable personal critique and judgement.

Here we see mistrust of the investigatory process itself. 'Pat' speculates that video evidence might slightly contradict remembered evidence, suggesting that a trivial inaccuracy might bring their credibility into question in court: "'*. . . you didn't do that at '52 [minutes past the hour], you did it at '53 [minutes past the hour], so how much of the rest of your statement is inaccurate*?*"*. 'Pat' appears to be concerned about being vilified and unfairly criticised in an adversarial investigation process.

There is a fear that their career would not survive such a scenario. 'Pat' anticipates being "*suspended*" from duty suggesting that "*It's a very ambulance thing, isn't it, a very paramedic thing that we associate discipline and error in the same bracket.*" This impacts how 'Pat' feels

about error making, with an emphasis on punishment rather than learning, creating additional fear and anxiety.

In the next example, 'Charlie' talks about what they anticipate happening from an investigatory point of view, should a patient with suspected ABD suffer a cardiac arrest in the context of an AP administering sedation.

> *"When you look at the blood gases of these patients and when you look at their physiology you just realise they are so, so sick some of them. But, we are going to go through that in detail, and we'll wheel out 'experts' who've never seen these patients in the community before [. . .] and we will be critiqued to that."*

> 'Charlie'

'Charlie' sheds additional insight into anxiety about the investigatory process. Specifically, what is articulated is mistrust in a process where people who lack APs experience of clinical practice in context, may be wheeled out as '*experts*'. There are two issues here. One is that the authority to critique is placed in the hands of these '*experts*' and that their lack of relevant experience may be detrimental to the outcome. The second is an implicit failure to recognise and validate APs' professional identities and their specialist knowledge and expertise.

Not all participants shared 'Pat' and 'Charlie's fear of the investigatory process. 'Glenn' below spoke positively about the support they would receive from their employer in such an event.

> *"I have real confidence that providing I do the right thing at the right time, that I will have a consultant doctor in the form of [a medical director] turn up after me in the court, and say 'I think that's about right and that's what we wanted people to do and this is why'. I wouldn't do this if I didn't feel that there was support behind it."*

> 'Glenn'

'Glenn' brings insight about the importance of employer assistance and support. Acknowledging that a serious event may occur where bold after-event employer support would be needed. 'Glenn' is confident that their employer would back them up '*in court*'. This confidence is predicated on 'Glenn' doing '*the right thing at the right time*', which is an important caveat given the ambiguity hampering diagnosis and treatment planning highlighted earlier. Our analysis cannot speak to why 'Glenn' experiences confidence where 'Pat' and 'Charlie' do not. It does, however, suggest the importance of employers creating a culture of high psychological safety in which the risks which APs and their patients face are understood, and support is made explicit.

**Needing to negotiate with other professionals.**   The needing to negotiate with other professionals theme comprised challenges experienced by APs when dealing with police and other ambulance clinicians at the scene of ABD incidents, and focussed on how participants experienced these interactions.

'Charlie', for example, attended a suspected ABD case in which the police are already at the scene and found their interactions to be straightforward in this case.

> *"I said I'm really worried this is ABD, and I'm really concerned about what might be [. . .] the cause of this, and the police officer went, who was the kind of lead officer until the Sergeant arrived subsequently, said to me 'no I think this is ABD, this is like exactly like the video.'"*

> 'Charlie'

In this incident, 'Charlie' connects with the police officer on their clinical concerns, developing a sense of urgency. The police officer validates 'Charlie's' clinical impression. Here, it appears that the sharing of information between professionals has facilitated cooperation, and enabled a helpful relationship based on a shared recognition of the situation, insofar as the patient is an ABD patient for the purposes of management.

In the next example, 'Eddie' recounts an incident they attended, in which a patient suspected of having ABD had been subdued by several police officers and was restrained physically, as well as mechanically with handcuffs and leg restraints. The participant suggested to the police that they reduce the level of physical restraint, after which the patient's agitation reduced.

> *"That was met with a little bit of resistance from the police. They had had to fight with him quite a lot to get him under control, and there was me saying, 'right, thanks for all your hard work officers, but now you can let go.' And there was some concern that obviously he may then become very agitated, and they would have to start from square one in terms of restraint."*

'Eddie'

'Eddie' acknowledges the events which have occurred prior to their arrival and understands the consequences for the police if reduction in restraint is unsuccessful–they will have to '*start from square one'*. The police hold an important gate-keeping role here and recognising that their responsibilities and concerns differ from those of APs appears to have been an important factor in fostering cooperation in this situation. 'Eddie' goes on to emphasise the importance of providing meaningful explanations to the police: "*This is why, and if it goes wrong, this is what we will do. So really explaining to police why I want them to do it, not just 'do it'."*, echoing the approaches described by other participants.

In the following scenario, 'Jordan' has arrived at an incident where a man who has taken a large amount of cocaine and is agitated and is subject to significant physical restraint. During the struggle to bring the man under control, a police officer has been injured and the police are explicit in their expectation that 'Jordan' will provide chemical sedation, however' Jordan' does not agree that the patient requires this intervention.

> *"He said 'we're not going to stop physically restraining them until you sedate him'. I was thinking I was being coached into chemical sedation and he says, 'we've had you guys out before, we've never had this problem before, just give him [. . .] sedation. I said listen, this is me, I'm deciding whether I'm going to sedate this patient. You're not telling me what to do and at the moment the only thing I think is causing this agitation is you physically restraining him."*

'Jordan'

As in the previous example, there is a dynamic which involves one professional group influencing the actions of another. In this case, it is the police attempting to influence the actions of APs, rather than the other way around as described earlier.

'Jordan' withholds chemical sedation because it is not clinically indicated. The police are seeking an alternative to physical restraint in a difficult case, perhaps highlighting the different lenses through which police officers and APs view these events, reinforcing the need for professionals to manage both agreement and disagreement on what course of action to take.

It seems that mutual sharing of information contributes to successful negotiations between professional groups. The establishment of professional connections between professional groups, which respond to different professional concerns and recognises the different ways police and APs view these events fosters cooperation.

**Recognising and establishing primacy of care.** The recognising and establishing primacy of care theme encompasses the participants' experiences of conceptualising the person with ABD as a patient and engaging with their professional duty of care for them. In this first example, 'Eddie' talks about the way police and APs as distinct professional groups view ABD patients.

> *"I think the police can sometimes see them as a bad person, whereas from a clinical perspective, I see these patients as very sick, and this is a medical emergency and actually this person doesn't really understand what they're doing. . ."*

'Eddie'

Here, 'Eddie' sheds light on the importance of how the person is categorised, specifically as a 'patient' with ABD rather than as a person who is characterised as '*bad*'. 'Eddie' emphasises the importance of keeping a focus on patients being '*critically unwell*' to avoid getting "*caught up in this escalation where the patient is getting more and more violent, 'cause then you, you can get angry at the patient. . .*". 'Eddie' is specifically concerned here about how conceptualising ABD patients as 'bad' rather than sick might affect how professionals engage with them. They highlight the potential for a vicious cycle of agitation, resistance, and restraint, and are worried that the patient's agitation will trigger stress and anger within themselves which might influence how they respond to the patient. The professional identity of police and APs again appears to have an important impact on how they interpret ABD patient presentations, and how they respond to them, and it appears that both professional groups have a need to maintain self-control when dealing with difficult cases involving ABD.

In the following example, 'Aubrey' describes an incident where police are also in attendance and have commenced restraining a person.

> *"I'm not a huge fan of spit hoods, I'd rather use an oxygen mask. You know and reminding the police that they're sick and not bad, and that they are patient not a criminal and you know, just creating that culture on scene."*

'Aubrey'

'Aubrey' volunteers a dislike of spit hoods (also known as spit guards), which are available to some UK police officers to protect them from detainees spitting on or at them. 'Aubrey' associates spit hoods with law enforcement. In this scenario, the symbolism of the oxygen mask suggests a clinical encounter, whereas the symbolism of a spit hood characterises a policing encounter.

'Aubrey' acknowledges that the police have developed their knowledge and approach to ABD patients and suggests that '*reminding*' police that agitation in the context of ABD is clinically rather than behaviourally driven is helpful. This again characterises the incident as clinical rather than a law enforcement issue and suggests that 'Aubrey' perceives that the behaviours of the police may veer towards traditional policing behaviours at times in these scenarios. This appears to necessitate action to bring others back to the shared understanding which will facilitate cooperation between police and APs at the scene. 'Aubrey' speaks of creating a clinical culture at the scene, which they indicate might be different than a policing

culture, but which facilitates the cooperation of the police and APs around a shared understanding of the person as patient first, and detainee second.

In the next example, 'Glenn' describes attending ABD incidents with the police and how they interact at the scene.

> *"It's a clinical problem, not a law enforcement problem so typically what happens is, the police in the more extreme cases will send a supervisor. So you find yourself that supervisor, you make sure that their body camera is turned on. You find a couple of ambulance staff. And you get a little huddle together and one of the very early things you say is this is now my problem, and I just need your help to sort the problem out [. . .] I find that once you once you use some very clear explicit language around that to say this is my problem, we can [manage] them under the capacity act, I'm responsible."*

'Glenn'

Here, 'Glenn' is positioning themselves as a leader and decision-maker. They establish ownership of the situation and ownership of the consequences. 'Glenn' is relieving the police of some responsibility, perhaps serving to alleviate some of their professional concerns regarding potential consequences should the patient have a poor outcome. 'Glenn' is presenting themselves as an enabler, in a context where police are concerned about death after police contact and the consequences that brings both in human terms for the patient, their families and the professionals involved, and in relation to the burdens of prolonged investigation. The adoption of a leadership position, taking responsibility for the scenario and clearly articulating what the 'ask' is of the police appears helpful in developing cooperation.

Like 'Charlie's' example above, 'Glenn' also grounds their decision-making in relevant legislation. This may serve to connect the clinical perspective of the APs with the legal lens of the police and addresses how the responders might justify restricting the patient's liberty in legal terms. 'Glenn' is connecting with the police on common ground, the use of the Mental Capacity Act (2005).

'Harper' touched on the issue of more junior paramedics and their interactions with the police.

> *"If you've got an inexperienced member of staff, they're more likely to be led by the police. [. . .] They may not even know that we should have primacy of care and we can actually say to the police, look you're restraining him wrong, you need to do it this way or however you want to phrase it. I suppose it goes wider to that societal thing where you always obey police. They were swayed by the police informing them that he was a violent patient, so to be careful, but actually at the time there was quite a lot of police officers restraining him. [. . .] They then just took that for granted even though they didn't do their own assessment."*

'Harper'

In discussing their observations of less experienced paramedics, 'Harper' makes several attributions in suggesting that they may be easily influenced by police officers. 'Harper' attributes a lack of appreciation of who should have primacy of care to inexperience and suggests that wider social norms about obeying the police may '*sway*' inexperienced clinicians against questioning them. At the point of cardiac arrest, the patient in this case became unambiguously a patient experiencing a clinical emergency which should be clinically led, with policing issues secondary. For 'Harper' however, the case was unambiguously a clinical emergency prior to this stage.

'Harper' sheds light on the importance of professional curiosity. In this specific case, 'Harper' postulates that the paramedics were influenced by the warnings of the police and as a result have not explored the severity of the patient's illness for themselves, and in doing so overlooked the need to intervene and establish clinical leadership of the incident. Noting this is an attribution, an alternative view might be that the paramedics were respecting the expertise of police in recognising and managing violence.

## Discussion

The purpose of this study was to explore the factors which influence APs when making restraint decisions for patients experiencing ABD, how they understand and manage these influences in practice and to explore the nature of any concerns for practitioners. This research was motivated by increasing professional interest in ABD, as well as new and emerging practice in this area relating to pharmacological interventions and a gap in the literature in relation to what influences decision-making in restraint situations.

Our findings echo the international literature highlighting the sense of vulnerability experienced by paramedics working in the ambulance setting in relation to decision-making [14, 18–23, 25, 26, 28]. ABD as a concept refers to agitation which may result from multiple causes and several differential diagnoses [3], this contributes to a complex and ambiguous decision-making context, which is complicated by a lack of information on which APs are required to make important risk-benefit decisions. This ambiguity appears closely linked with concerns about fair and balanced investigation in the event of a patient deteriorating whilst in AP care, and the degree to which clinicians expect their employers to support them after an adverse event. There was a sense that participants did not always feel trusted or respected as professionals, echoing research that finds paramedics perceive a culture of discipline and blame within UK ambulance services [24]. Whilst our research attested to anxieties APs feel in relation to investigatory processes, not all participants felt this way with some expressing trust and confidence in employer support. Further research may assist in understanding what might contribute to these different perspectives on trust and psychological safety in the workplace.

We found that key influences on decision-making in ABD arose from the social and professional relationships between responders at ABD incident scenes. Thus, our understanding of decision-making needs to shift away from the notion of an atomised individual-level process. The sharing of information and validation of each other's professional expertise was an important factor in successful joint working and reduced the complexity and ambiguity of incidents. Recognising the different professional goals and concerns of police and APs, and the expression of mutual respect was relevant in APs being able to decide to reduce physical restraint. The relationships between APs and the police identifying and categorising people presenting with ABD as patients rather than perpetrators was also relevant. Participants took active steps to situate incidents as healthcare encounters. This context was key to locating ABD incidents in the span of control of APs within power gradients at scene where it was essential for APs to establish ownership of management decisions which were clinical in nature, including and especially those relating to restraint. One potential contributing factor is how clinicians navigate the traditional social authority and expertise in managing violence and aggression which the police hold, with their duty of care as healthcare professionals. This may be a particular issue where the police have arrived first and defined the context of the scene prior to the arrival of APs. This may be of particular importance in the context of ABD where the maintenance of law and order and providing medical care are, to an extent, entwined. This sets the scene for negotiations with police, predominantly in relation to reduction or cessation of physical

restraint, or the initiation of chemical sedation, all with the goal of increasing safety for both the patient and the responders and to facilitate safe onward transfer and clinical care.

APs are experienced paramedics with extended education and training which includes preparation for attending ABD patients. APs are specifically prepared to be 'the clinical lead' in the cases they attend and wear different coloured epaulettes to assist other responders in identifying them at the scene of incidents. It might be reasonable to expect a newly qualified paramedic to be less confident in challenging another professional group with relative social authority and expertise in managing agitation and restraint, and perhaps be less likely to effectively establish primacy of care where leadership or the 'ownership' of an incident is contested or unclear. APs negotiated clinical leadership and primacy of care at ABD incidents by fostering a shared understanding of the nature of the problem and its associated risks with other professionals at the scene. As was the case in studies relating to restraint in the in-patient setting [32], as experienced practitioners APs were comfortable trying less 'hands on' techniques. Chemical sedation appeared to be favoured by police over physical restraint, and this led at times to APs feeling 'coached' into providing a pharmacological intervention and tensions arose where APs determined it was not a clinically appropriate course of action. This perhaps highlights the different lenses through which police and APs view chemical sedation as an intervention.

## Implications for practice

This study adds to the evidence base which highlights a sense of professional vulnerability experienced by paramedics working in the ambulance setting in relation to high-risk decision-making. This may have negative effects on the health and wellbeing of staff, and is a patient safety issue [49]. More cultural change is needed in ambulance services to improve psychological safety to address this.

Our analysis calls attention to the importance of recognising the role of context and social relations in shaping how decisions are made in the pre-hospital ambulance setting. Failure to consider how decisions are made in the context of these social relationships may negatively influence patient safety. In preparing for clinical practice, issues around inter-professional negotiation, assertiveness and primacy of care represent important opportunities for development in pre and post qualification education and training. This is of particular importance in preparing APs for the complexities and risk associated with chemical sedation. A focus on inter-professional training with the police, alongside high-fidelity training and observation may be helpful interventions in achieving this.

## Limitations and opportunities for future research

This study took place in a single setting, which is described here to enhance transferability and is not intended to be generalisable to every ambulance service. The LAS employs a small group of around 40 APs who staff four response cars on a 24/7 basis with targeted deployment to suspected ABD incidents, seeing 237 in a year according to one study [13]. The exposure of ambulance clinicians to suspected ABD cases may vary between different ambulance services.

This was a small study in nomothetic, statistical terms, but not in idiographic terms [50] where the focus was understanding experiences and sense-making within a small community of professionals. More critical for evidentiary adequacy in this study is adequate discrepant case analysis and adequate variety in kinds of evidence [51]. In relation to the first, participants were self-selected volunteers, and this may reflect a personal interest in ABD which may affect the data collected. In relation to the second, a single data collection technique was used. Although the interviews were in-depth and produced a rich data corpus, it is recognised that

the inclusion of observational techniques would have enhanced the robustness of the study's findings, by providing an additional source of evidence and reducing reliance on participants recalling their experiences. Ethnographic studies of the full decision-making process from emergency call to handover of care may shed further light on AP decision-making.

This study only included APs, therefore the perspectives of other professionals involved in ABD restraint decisions were not available. Studies exploring the perspectives of general paramedics and EMTs who are also involved in managing ABD patients at incidents scenes, as well as bystanders and emergency department clinicians who receive ABD patients would bring important perspectives. It is also recognised that the perspectives and priorities of police officers are important factors and form part of the context in which APs make restraint decisions, this is also an important opportunity for future research.

## Conclusion

This study provides new and in-depth understanding of the decisions made by APs in relation to ABD patients. This is particularly important given recent concerns about death in the context of restraint [52, 53]. Our results highlight the uncertainty which surround ABD decisions, increasing their risk, and indicates that the ability to navigate this uncertainty will increase the safety of these decisions in practice.

We found that the decisions APs make in relation to ABD patients are complex and are influenced by several different factors. A key influence was the need to characterise incidents as medical, which participants used as an enabler to establish leadership and decision-making control. APs focused on de-escalation techniques and sought to reduce physical restraint where possible, intervening with pharmacological interventions if necessary to achieve this. The social relationships and interactions with patients and other professionals at the scene were key to successful outcomes and formed an important context for decision-making. We determined that restraint decisions in ABD incidents are a cause of concern for some APs. A lack of psychological safety was also important, with fears of professional detriment accompanying a poor patient outcome.

## Acknowledgments

We would like to thank all our study participants for their time and contributions.

## Author Contributions

**Conceptualization:** Jaqualine Lindridge, Timothy Edwards, Leda Blackwood.

**Formal analysis:** Jaqualine Lindridge, Timothy Edwards, Leda Blackwood.

**Investigation:** Jaqualine Lindridge.

**Methodology:** Jaqualine Lindridge, Leda Blackwood.

**Project administration:** Jaqualine Lindridge.

**Supervision:** Timothy Edwards, Leda Blackwood.

**Writing – original draft:** Jaqualine Lindridge.

**Writing – review & editing:** Jaqualine Lindridge, Timothy Edwards, Leda Blackwood.

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
