## [Decision Letter · Decision Letter 0]

11 Sep 2023

PONE-D-23-16883'This is on me.' A qualitative exploration of restraint decisions made by advanced paramedics in the UK when managing acute behavioural disturbance (ABD).PLOS ONE

Dear Dr. Lindridge,

Thank you for submitting your manuscript to PLOS ONE. After careful consideration, we feel that it has merit but does not fully meet PLOS ONE’s publication criteria as it currently stands. Therefore, we invite you to submit a revised version of the manuscript that addresses the points raised during the review process.

We look forward to receiving your revised manuscript.

Kind regards,

Bidisha Banerjee, Ph.D.

Academic Editor

PLOS ONE

Journal Requirements:

2. We noted in your submission details that a portion of your manuscript may have been presented or published elsewhere:

"An abstract reporting the results of this study is due to be published in the Emergency Medicine Journal as part of the conference proceedings for the 999EMS Research Forum, following a poster presentation at that forum."

Please clarify whether this conference proceeding or publication was peer-reviewed and formally published. If this work was previously peer-reviewed and published, in the cover letter please provide the reason that this work does not constitute dual publication and should be included in the current manuscript.

**Additional Editor Comments:**

Comments are mentioned at the end of the email. 

Reviewers' comments:

Reviewer's Responses to Questions

**Comments to the Author**

1. Is the manuscript technically sound, and do the data support the conclusions?

Reviewer #1: Yes

Reviewer #2: Partly

2. Has the statistical analysis been performed appropriately and rigorously? 

Reviewer #1: N/A

Reviewer #2: No

3. Have the authors made all data underlying the findings in their manuscript fully available?

Reviewer #1: No

Reviewer #2: No

4. Is the manuscript presented in an intelligible fashion and written in standard English?

Reviewer #1: Yes

Reviewer #2: No

5. Review Comments to the Author

Reviewer #1: Thank you for meaningful and insightful work. May i provide some suggestions for your considerations.

The Title starting with "That is on me." It could be leading the readers to seme aspects of the APs' perceptions or psychological impact, etc. The statement could not reflect or conclude all the themes developed. Therefore, suggest removing it.

Introduction and background: overall, the general sensing for the background is that some inferences allude there are needs or gaps to be addressed for the topic of APs' decision-making in restraints for people with ABD. However, the big proportion of the literature review were discussing concerns of paramedics, not APs, and in management of other conditions than ABD. Hence, the relevancy of the review could not very convincing. Suggest relooking into adding more specific background of APs, your targeted population, in the specific group of ABD.

Line 75, "Research has identified that paramedics have felt unprepared by their formal training", what about the APs? was it due to ineffective training?

Line 82, other heathcare professionals also rely on tacit knowledge when making restraint decisions." Is it a common problem in all professionals handling restraint. so, what is unique need to look into the APs?

Line 84, since there was paucity of research, how did you gather there are concerns/gaps to address in the APs' restraint process?

There might be a need to define "clinician", as it might mean different scope of role in some countries? are they doctors?

Are paramedics and APs used interchangeably in the manuscripts? As your target population is APs, Line 84-125, concerns of paramedics were discussed. Do APs face similar issue?

Philosophy: the adoption of critical realist combines ontological realism and epistemological constructivism is a great application in this topic. To facilitate better understanding by a wider group of readers, it would be great to elaborate on how its applied.

Sampling: seven out of ten volunteers were selected. why the three are excluded? will be any chance the insider researcher could be asking leading questions? how is it avoided?

Data collection and processing: how the informal member checking is sone? author TE did not have access to information that could identify the participants. what is the implication of this?

Analysis: Line 393 "psychological safety" might not be very suitable to use here, as it could be caused by a range of factors. suggest rephrasing it, e.g. impact of employer support, or organizational support.

Reviewer #2: In the current study, seven semi-structure in-depth interviews have been reported and authors have used Braun and Clarke’s suggestion as their argument for having small sample size. However, it has not been justified properly.

Therefore, it is suggested to collect more data or present sufficient rationale for choosing such small sample size.

Please also provide demographic information of your sample.

In the manuscript, many technical terms have been used but they have not been defined properly and restructuring some sentences. The manuscript requires language revisions to improve readability and meaning of the text.

The rationale of the study is mentioned but for better structure and organization, please provide rationale of the study as a separate subsection.

The term ‘Philosophy’ is an ambiguous term. Please replace it with appropriate sub-heading.

It will be better if you could give the demographic info (age and sex) of the respondents along with their excerpts.

The result section seems confusing. It should not be named ‘analysis’ since this section talks about the findings. Moreover, it should be restructured for more clarity. It seems that themes are given within the context of participants. It should be made clear from the beginning.

6. PLOS authors have the option to publish the peer review history of their article (what does this mean?). If published, this will include your full peer review and any attached files.

Reviewer #1: **Yes: **Li Ziqiang

Reviewer #2: No

---

## [Author Response · Author response to Decision Letter 0]

21 Oct 2023

8th October 2023

Reference: PONE-D-23-16883

'This is on me.' A qualitative exploration of restraint decisions made by advanced paramedics in the UK when managing acute behavioural disturbance (ABD).

PLOS ONE 

Dear Dr. Banerjee,

Thank you for your, and the reviewers’ feedback and the invitation to submit a revised version of our manuscript.

We have revised the paper and are pleased to re-submit this for your consideration. We outline our response to your, and the reviewers, points below.

Editor’s Comments

We have named the 3 updated files in accordance with PLOS ONE’s guidelines and the instructions in the revision letter. We have also checked through the manuscript to ensure it meets PLOS ONE requirements.

We noted in your submission details that a portion of your manuscript may have been presented or published elsewhere. Please clarify whether this conference proceeding or publication was peer-reviewed and formally published. If this work was previously peer-reviewed and published, in the cover letter please provide the reason that this work does not constitute dual publication and should be included in the current manuscript.

The draft abstract and poster to which we refer have not been and will not be subject to peer review. They have been submitted to the annual 999 Emergency Medical Services research conference, and the poster was displayed during that conference. The abstract has been omitted from publication of the proceedings.

Reviewer’s Comments

1. Is the manuscript technically sound, and do the data support the conclusions?

The manuscript must describe a technically sound piece of scientific research with data that supports the conclusions. Experiments must have been conducted rigorously, with appropriate controls, replication, and sample sizes. The conclusions must be drawn appropriately based on the data presented. Reviewer #1: Yes Reviewer #2: Partly

We have responded to individual reviewer comments below.

2. Has the statistical analysis been performed appropriately and rigorously? Reviewer #1: N/A. Reviewer #2: No

This is qualitative research and there is no statistical analysis. We trust that no further comment on statistical analysis is required.

3. Have the authors made all data underlying the findings in their manuscript fully available? The PLOS Data policy requires authors to make all data underlying the findings described in their manuscript fully available without restriction, with rare exception (please refer to the Data Availability Statement in the manuscript PDF file). The data should be provided as part of the manuscript or its supporting information, or deposited to a public repository. For example, in addition to summary statistics, the data points behind means, medians and variance measures should be available. If there are restrictions on publicly sharing data—e.g. participant privacy or use of data from a third party—those must be specified. Reviewer #1: No Reviewer #2: No

Anonymised transcripts are available via the University of Bath Research Data Archive. The data relates to a small professional group and a controversial subject. The data is available to researchers on application, rather than on an open access basis. This is to protect the privacy of participants and this requirement forms part of the ethical approval for this study. The reference and DOI for the data will is as follows: Lindridge, J., Edwards, T., Blackwood, L., in press. Dataset for "Advanced paramedics' restraint decision-making when managing acute behavioural disturbance (ABD): A UK based qualitative investigation.". Bath: University of Bath Research Data Archive. https://doi.org/10.15125/BATH-01326. The DOI will be activated on publication of the manuscript.

4. Is the manuscript presented in an intelligible fashion and written in standard English?

PLOS ONE does not copyedit accepted manuscripts, so the language in submitted articles must be clear, correct, and unambiguous. Any typographical or grammatical errors should be corrected at revision, so please note any specific errors here. Reviewer #1: Yes Reviewer #2: No

We have responded to individual reviewer comments below.

5. Review Comments to the Author

Reviewer #1: Thank you for meaningful and insightful work. May i provide some suggestions for your considerations.

The Title starting with "That is on me." It could be leading the readers to seme aspects of the APs' perceptions or psychological impact, etc. The statement could not reflect or conclude all the themes developed. Therefore, suggest removing it.

We would like to thank Reviewer 1 for their engagement with our research and for their positive comments. We also appreciate the guidance they have provided. We agree that the title might be misleading and thank the reviewer for bringing this to our attention. We have now revised the title to: Advanced paramedics' restraint decision-making when managing acute behavioural disturbance (ABD): A UK based qualitative investigation. 

Introduction and background: overall, the general sensing for the background is that some inferences allude there are needs or gaps to be addressed for the topic of APs' decision-making in restraints for people with ABD. However, the big proportion of the literature review were discussing concerns of paramedics, not APs, and in management of other conditions than ABD. Hence, the relevancy of the review could not very convincing. Suggest relooking into adding more specific background of APs, your targeted population, in the specific group of ABD.

We are grateful for this feedback. There is very little literature on APs and accordingly we have needed to draw on wider literature which we believe does have relevance. We agree this was not clear in the previous manuscript; a paragraph outlining the rationale for including literature relating to paramedics in the literature review has been included to make the relevance of this research included much clearer. We have also now specifically described the relevance of evidence cited to APs, we hope this addresses the reviewer’s concerns.

Line 75, "Research has identified that paramedics have felt unprepared by their formal training", what about the APs? was it due to ineffective training?

As noted above, there is little literature on AP’s experiences so we cannot comment on the extent to which this is also true for this group. We have amended the manuscript to make this clearer, and are grateful to the reviewer for bringing this to our attention.

Line 82, other heathcare professionals also rely on tacit knowledge when making restraint decisions." Is it a common problem in all professionals handling restraint. so, what is unique need to look into the APs?

Our literature review did not explicitly explore whether relying on tacit knowledge is common to all or most healthcare professionals. The relevance to looking specifically at APs is related to their extended scope of practice, which includes forms of chemical restraint. We have amended the manuscript to make this clearer, we are grateful for the feedback.

Line 84, since there was paucity of research, how did you gather there are concerns/gaps to address in the APs' restraint process?

We thank reviewer 1 for your highlighting this omission. We determined that there were concerns and gaps to address because this area has been highlighted as a matter of concern by coroners, regulators and the professional body for paramedics in the UK. We have included more information in the introduction to explain this, as well as highlighting a scoping review into patient safety in ambulance service which also identified the need for more research in this area.

There might be a need to define "clinician", as it might mean different scope of role in some countries? are they doctors?

We are grateful to reviewer 1 for highlighting this, and we agree that the definition of clinician is likely to vary internationally. We have added a definition so it is clear to readers how we are interpreting this term and the types of healthcare worker it relates to in the UK pre-hospital ambulance setting.

Are paramedics and APs used interchangeably in the manuscripts? As your target population is APs, Line 84-125, concerns of paramedics were discussed. Do APs face similar issue?

We agree a clearer distinction between paramedics and APs would be helpful and thank reviewer 1 for bringing this to our attention. The relevance of paramedic research on APs has been made clearer throughout this section, with explicit explanations on how literature is applicable to APs where appropriate. We hope this has improved the manuscript and addresses these concerns.

Philosophy: the adoption of critical realist combines ontological realism and epistemological constructivism is a great application in this topic. To facilitate better understanding by a wider group of readers, it would be great to elaborate on how its applied.

We thank reviewer 1 for the feedback. Further detail relating to critical realism has been added to this section to explain how we have applied this philosophy to the phenomenon under study, in that we take the position that the decisions made in this context are real and our participants make sense of that reality in a social constructed way. We hope this has enriched the manuscript.

Sampling: seven out of ten volunteers were selected. why the three are excluded? will be any chance the insider researcher could be asking leading questions? how is it avoided?

In relation to sampling, the 3 potential participants were not specifically excluded but rather did not follow up on their expression of interest. Accordingly, we have removed reference to these three. In relation to questioning, participants were invited to tell stories and to give accounts of events that were meaningful to them. This was done to ensure that the participant could raise what they thought was important. The interviewer was aware of her insider status and the risks involved, she addressed this through open ended questions and member checking. We have provided additional information on how the interview was approached which we hope will address the reviewer’s concerns. 

Data collection and processing: how the informal member checking is sone? author TE did not have access to information that could identify the participants. what is the implication of this?

In relation to member checking, this was achieved by the interviewer restating points in the interview and inviting the participants to confirm accuracy of understanding. A note to explain this has been added. In relation to TE's access to participant personal data, only one supervisor (the academic supervisor) required access to participant personal data. This has been clarified in the text. Access to participant personal data was necessary for TE’s role as practice-based supervisor, and there is no implication of this in relation to data analysis. 

Analysis: Line 393 "psychological safety" might not be very suitable to use here, as it could be caused by a range of factors. suggest rephrasing it, e.g. impact of employer support, or organizational support.

We thank reviewer 1 for the feedback. We have amended the manuscript accordingly, replacing ‘psychological safety’ with ‘employer assistance and support.

Reviewer #2: In the current study, seven semi-structure in-depth interviews have been reported and authors have used Braun and Clarke’s suggestion as their argument for having small sample size. However, it has not been justified properly. Therefore, it is suggested to collect more data or present sufficient rationale for choosing such small sample size.

We are grateful to reviewer 2 for this feedback and agree that this point would benefit from additional justification. We have amended the sampling and recruitment section to include more information on our rationale for determining the final sample size. We determined that the participants had a high level of experience of the phenomenon under study, and the interview dialogue was of high quality. Consequently, we agreed that seven interviews generated sufficient rich and complex data to generate meaningful theme. We trust this addresses reviewer 2’s concerns.

Please also provide demographic information of your sample.

This is provided in the 'participants' section, this has been augmented to note all of the participants are of adult working age.

In the manuscript, many technical terms have been used but they have not been defined properly and restructuring some sentences. The manuscript requires language revisions to improve readability and meaning of the text.

We thank reviewer 2 for this feedback. We have reviewed the manuscript and clarified technical terms throughout, as well as reviewing language and syntax throughout, making revisions throughout.

The rationale of the study is mentioned but for better structure and organization, please provide rationale of the study as a separate subsection.

We thank reviewer 2 for their comments and agree the structure and organisation of the manuscript could be improved. We have added a ‘research rationale’ subsection accordingly.

The term ‘Philosophy’ is an ambiguous term. Please replace it with appropriate sub-heading.

We thank reviewer 2 for their comments and have amended this sub-heading to 'research philosophy'.

It will be better if you could give the demographic info (age and sex) of the respondents along with their excerpts.

The demographics of participants are provided in a separate section. This section explains that of the seven participants, only one was female and highlights that to protect anonymity, gender-neutral pseudonyms were used for all participants. At the time of the study, the majority of APs in the population were male, and the sample represented approximately 18% of the total population, therefore there is a risk of inadvertently identifying participants without careful anonymisation. The research team remain of the view that including the gender of participants along with their excerpts represents a risk to participant anonymity.

The result section seems confusing. It should not be named ‘analysis’ since this section talks about the findings. Moreover, it should be restructured for more clarity. It seems that themes are given within the context of participants. It should be made clear from the beginning.

We thank reviewer 2 for their feedback. We have re-titled the analysis section 'findings'. In relation to the structure of this section, this is organised as a thematic analysis and therefore consists of a balance of data extracts from participants and analytic narrative. A explanation of this has been added at the beginning of this section which is intended to assist the reader navigate the analysis which follows, and make this clearer. We hope this addresses the reviewer’s concerns.

In conclusion, we hope you find our amendments to be satisfactory, and we look forward to hearing from you.

Kind regards

Ms Jaqualine Lindridge

---

## [Decision Letter · Decision Letter 1]

1 Feb 2024

PONE-D-23-16883R1Advanced paramedics' restraint decision-making when managing acute behavioural disturbance (ABD): A UK based qualitative investigation.PLOS ONE

Dear Dr. Lindridge,

Thank you for submitting your manuscript to PLOS ONE. After careful consideration, we feel that it has merit but does not fully meet PLOS ONE’s publication criteria as it currently stands. Therefore, we invite you to submit a revised version of the manuscript that addresses the points raised during the review process.

We look forward to receiving your revised manuscript.

Kind regards,

Peivand Bastani

Academic Editor

PLOS ONE

Reviewers' comments:

Reviewer's Responses to Questions

**Comments to the Author**

1. If the authors have adequately addressed your comments raised in a previous round of review and you feel that this manuscript is now acceptable for publication, you may indicate that here to bypass the “Comments to the Author” section, enter your conflict of interest statement in the “Confidential to Editor” section, and submit your "Accept" recommendation.

Reviewer #3: (No Response)

2. Is the manuscript technically sound, and do the data support the conclusions?

Reviewer #3: Yes

3. Has the statistical analysis been performed appropriately and rigorously? 

Reviewer #3: N/A

4. Have the authors made all data underlying the findings in their manuscript fully available?

Reviewer #3: No

5. Is the manuscript presented in an intelligible fashion and written in standard English?

Reviewer #3: No

6. Review Comments to the Author

Reviewer #3: The researcher explored restraint decision-making by advanced paramedics(APs) when managing acute behavioural disturbance (ABD) through four themes, it is an interesting study and the findings are useful as a guide for restraint decision-making when managing ABD. Although it is a qualitative study, a useful attempt. There are a few areas for consideration and potential improvements：

1. The context of this study targeted ABD incidents in the UK pre-hospital ambulance setting and it is hoped that reflecting this context in the title, will help the reader to clarify the topic of the study. Because in some countries and regions, the reference to paramedics may not be immediately associated with pre-hospital ambulance service.

2. line 18, the aim of the study in the abstract should also highlight that the study setting is ABD incidents in the UK pre-hospital ambulance setting.

3. Line 28, hoping to show that the themes are summarized from the interviews.

4. A larger proportion of the literature review refers to paramedics, with less mention of Aps in this study, and the authors mentioned fewer Aps-related studies in the previous peer review, but an appropriate increase in the content of the literature review to include a description of Aps would be beneficial.

5. The content after the abstract and before the methods is very similar to the introduction. The authors may have written in points for the purpose of a better structure. But as far as the content seems confusing. It is recommended that the Introduction and background, Definitions, Literature review, and Research rationale be integrated into a new Introduction and background. When integrating, if the writer tends to write in points, then it is recommended that Definitions, Literature review, and Research rationale be categorized under the general heading of Introduction and background, which may be beneficial for readability.

6. In line 86, advanced paramedics shorthand APs have been provided in the text, but there are a large number of advanced paramedics in the text (e.g. lines 90, 100, 137).

7. Line 234, Paramedics (including APs), recommendation to delete Paramedics.

8. The sample size is indeed a concern, and it is worth noting that the authors have responded to this in the previous peer review, but there is still some concern about the sample size, and it is hoped that pointed out in the limitations.

9. The manuscript still has paramedics and APs used interchangeably, which seems confusing.

7. PLOS authors have the option to publish the peer review history of their article (what does this mean?). If published, this will include your full peer review and any attached files.

Reviewer #3: No

---

## [Author Response · Author response to Decision Letter 1]

17 Feb 2024

Dear Dr. Bastani,

Thank you for your, and the reviewers’ feedback and the invitation to submit a revised version of our manuscript.

We have revised the paper and are pleased to re-submit this for your consideration. We outline our response to your, and the reviewers, points below.

Reviewer’s Comments

4. Have the authors made all data underlying the findings in their manuscript fully available? The PLOS Data policy requires authors to make all data underlying the findings described in their manuscript fully available without restriction, with rare exception (please refer to the Data Availability Statement in the manuscript PDF file). The data should be provided as part of the manuscript or its supporting information, or deposited to a public repository. For example, in addition to summary statistics, the data points behind means, medians and variance measures should be available. If there are restrictions on publicly sharing data—e.g. participant privacy or use of data from a third party—those must be specified. Reviewer #3: No 

Anonymised transcripts are available via the University of Bath Research Data Archive. The data relates to a small professional group and a controversial subject. The data is available to researchers on application, rather than on an open access basis. This is to protect the privacy of participants and this requirement forms part of the ethical approval for this study. The reference and DOI for the data will be as follows: Lindridge, J., Edwards, T., Blackwood, L., in press. Dataset for "Advanced paramedics' restraint decision-making when managing acute behavioural disturbance (ABD): A UK based qualitative investigation.". Bath: University of Bath Research Data Archive. https://doi.org/10.15125/BATH-01326. The DOI will be activated on publication of the manuscript.

5. Is the manuscript presented in an intelligible fashion and written in standard English?

PLOS ONE does not copyedit accepted manuscripts, so the language in submitted articles must be clear, correct, and unambiguous. Any typographical or grammatical errors should be corrected at revision, so please note any specific errors here. Reviewer #3: No

We have responded to individual reviewer comments below.

6. Review Comments to the Author

We would like to thank Reviewer 3 for their engagement with our research and for their time in preparing this helpful review. 

1. The context of this study targeted ABD incidents in the UK pre-hospital ambulance setting and it is hoped that reflecting this context in the title, will help the reader to clarify the topic of the study. Because in some countries and regions, the reference to paramedics may not be immediately associated with pre-hospital ambulance service.

Thank you for alerting us to this, we agree and have amended the title, and short title, accordingly. The title now reads ‘Advanced paramedics’ restraint decision-making when managing acute behavioural disturbance (ABD) in the UK pre-hospital ambulance setting: A qualitative investigation.’

2. line 18, the aim of the study in the abstract should also highlight that the study setting is ABD incidents in the UK pre-hospital ambulance setting.

Thank you for highlighting this omission, this has been amended accordingly at line 22 with ‘The aim of this research is to explore the decisions made by APs when managing restraint in the context of ABD, in the UK pre-hospital ambulance setting.’ 

3. Line 28, hoping to show that the themes are summarized from the interviews.

We have clarified in the abstract that the themes arose from the interview data, now at line 27.

4. A larger proportion of the literature review refers to paramedics, with less mention of Aps in this study, and the authors mentioned fewer Aps-related studies in the previous peer review, but an appropriate increase in the content of the literature review to include a description of Aps would be beneficial.

Thank you for highlighting the need for further work on this section. We included a statement at line 130 to make it clearer that no studies specifically relating to APs were identified. We have also included an acknowledgement of several included studies which reported data from specialist or enhanced paramedics in addition to general paramedics (line 101). Whilst not working at an advanced level, specialist/ enhanced paramedics typically hold additional qualifications and work at a more enhanced scope of practice than general paramedics. We have also moved and refined the definition of paramedics and APs from earlier in the manuscript to line 69 in response to this feedback. We hope this addresses your concern and improves the readability of the manuscript.

5. The content after the abstract and before the methods is very similar to the introduction. The authors may have written in points for the purpose of a better structure. But as far as the content seems confusing. It is recommended that the Introduction and background, Definitions, Literature review, and Research rationale be integrated into a new Introduction and background. When integrating, if the writer tends to write in points, then it is recommended that Definitions, Literature review, and Research rationale be categorized under the general heading of Introduction and background, which may be beneficial for readability.

Thank you for highlighting this issue. We agree the structure could be improved. To address this, we have redrafted the content between the abstract and methods section (lines 46-144). We have removed the additional headings (definitions, literature review and research rationale) and the content is now presented in a single ‘Introduction and methods’ section. We have removed repetition, and integrated the content into a single section, which is slightly shorter as advised. We hope this improves the readability of the manuscript and satisfactorily addresses your feedback.

6. In line 86, advanced paramedics shorthand APs have been provided in the text, but there are a large number of advanced paramedics in the text (e.g. lines 90, 100, 137).

Thank you for highlighting this. We have reviewed the entire manuscript and made several corrections. The shorthand ‘AP’ is now used consistently throughout and is written in full only on its first use (line 58).

7. Line 234, Paramedics (including APs), recommendation to delete Paramedics.

Thank for highlighting this, we agree it is unhelpful to refer to both paramedics and APs here and have deleted ‘paramedics’ as recommended. Now at line 196.

8. The sample size is indeed a concern, and it is worth noting that the authors have responded to this in the previous peer review, but there is still some concern about the sample size, and it is hoped that pointed out in the limitations.

Thank you for highlighting this and acknowledging our response to previous peer review. We have now added a sentence in the limitations section at line 675: ‘This was a small study in nomothetic, statistical terms, but not in idiographic terms where the focus was understanding experiences and sense-making within a small community of professionals’. We have also provided a reference to a recent systematic analysis of justifications for sample size. 

9. The manuscript still has paramedics and APs used interchangeably, which seems confusing.

We have reviewed the entire manuscript to identify occasions where paramedics and APs were used interchangeably and made corrections throughout, particularly in our analysis where we agree references to ‘paramedics’ when analysing the data contributed by APs was confusing. The term ‘paramedic’ is now used much less frequently and much more specifically throughout the manuscript. Thank you for drawing this matter to our attention. 

In conclusion, we hope you find our amendments to be satisfactory, and we look forward to hearing from you.

Kind regards

Ms Jaqualine Lindridge

---

## [Decision Letter · Decision Letter 2]

8 Apr 2024

Advanced paramedics' restraint decision-making when managing acute behavioural disturbance (ABD) in the UK pre-hospital ambulance setting: A qualitative investigation.

PONE-D-23-16883R2

Dear Dr. Lindridge,

We’re pleased to inform you that your manuscript has been judged scientifically suitable for publication and will be formally accepted for publication once it meets all outstanding technical requirements.

Kind regards,

Peivand Bastani

Academic Editor

PLOS ONE

Additional Editor Comments (optional):

Reviewers' comments:

Reviewer's Responses to Questions

**Comments to the Author**

1. If the authors have adequately addressed your comments raised in a previous round of review and you feel that this manuscript is now acceptable for publication, you may indicate that here to bypass the “Comments to the Author” section, enter your conflict of interest statement in the “Confidential to Editor” section, and submit your "Accept" recommendation.

Reviewer #2: All comments have been addressed

Reviewer #3: (No Response)

2. Is the manuscript technically sound, and do the data support the conclusions?

Reviewer #2: Yes

Reviewer #3: (No Response)

3. Has the statistical analysis been performed appropriately and rigorously? 

Reviewer #2: Yes

Reviewer #3: (No Response)

4. Have the authors made all data underlying the findings in their manuscript fully available?

Reviewer #2: Yes

Reviewer #3: (No Response)

5. Is the manuscript presented in an intelligible fashion and written in standard English?

Reviewer #2: Yes

Reviewer #3: (No Response)

6. Review Comments to the Author

Reviewer #2: I thank to authors for revising the manuscript. The author(s) have included all recommendations which have improved the quality of the MS.

Reviewer #3: (No Response)

7. PLOS authors have the option to publish the peer review history of their article (what does this mean?). If published, this will include your full peer review and any attached files.

Reviewer #2: No

Reviewer #3: No
